# A combined variational and diagrammatic quantum Monte Carlo approach to the many-electron problem

Kun Chen [1] & Kristjan Haule [1]

Two of the most influential ideas developed by Richard Feynman are the Feynman diagram technique and his variational approach. Here we show that combining both, and introducing a diagrammatic quantum Monte Carlo method, results in a powerful and accurate solver to the generic solid state problem, in which a macroscopic number of electrons interact by the long range Coulomb repulsion. We apply it to the quintessential problem of solid state, the uniform electron gas, which is at the heart of the density functional theory success in describing real materials, yet it has not been adequately solved for over 90 years. Our method allows us to calculate numerically exact momentum and frequency resolved spin and charge response functions. This method can be applied to a number of moderately interacting electron systems, including models of realistic metallic and semiconducting solids.

[1] Department of Physics and Astronomy, Rutgers University, Piscataway, NJ 08854, USA. Correspondence and requests for materials should be addressed to K.C. (email: chenkun0228@gmail.com) or to K.H. (email: haule@physics.rutgers.edu)

 1

The success of the Feynman's diagram technique[1] rests on two pillars, the quality of the chosen starting point and one's ability to compute the contributions of high-enough order, so that the sum ultimately can be extrapolated to the infinite order.

Here we address the former by introducing the variationally optimized starting point, as discussed below, and we solve the latter by developing a powerful Monte Carlo method which can sum factorially large number of diagrams while massively reducing the fermionic sign problem by organizing Feynman diagrams into sign-blessed groups. The resulting Variational Diagramatic Monte Carlo (VDMC) method is a generic many-body solver, which is here tested on the classic solid state problem. We compute the spin and the charge response functions, which are directly accessed by the experiment, but remain challenging to compute by other methods. The accuracy of the calculated response functions is sufficiently high, so as to uncover previously missed fine structure in these responses.

## Results

**The optimized starting point Lagrangian.** In the Feynman diagrammatic approach, one splits the Lagrangian of a system, $L$, into a solvable part $L_0$ and the interaction $\Delta L = L - L_0$. The effect of the interaction is included with a power expansion in $\Delta L$, constructed using the Feynman diagrams. Such diagrammatic series achieves the most rapid convergence when the leading term $L_0$ captures the emergent collective behavior of the system, which can be very different from the non-interacting problem[2]. In the metallic state, which is of special interest in this paper, the low-temperature physics is described by the emergent quasiparticles interacting with a screened Coulomb interaction. We build an effective Feynman diagrammatic approach by explicitly encoding such physics in $L_0$. We screen the interaction in $L_0$ with a screening parameter $\lambda$, rendering the Coulomb interaction short-ranged $(V(r) \propto \exp(-r\sqrt{\lambda})/r)$. Correspondingly, a $\lambda$ counter-term must be added to $\Delta L$ to capture the non-local effects of the Coulomb interaction with high-order diagrams (see the Methods section). Similarly, we introduce an electron potential $\nu_{\mathbf{k}}$ which properly renormalizes the electron dispersion and also fixes the Fermi surface of $L_0$ to the exact physical volume, which is enforced by the Luttinger's theorem[3] (see the Methods section). In our simulations, such choice shows the most rapid and uniform convergence of the response functions for both small and large momenta.

Motivated by Feynman's variational approach[4], we take the screening parameter $\lambda$ as variational parameter which should be optimized to accelerate the rate of convergence. It was shown in the development of optimized perturbation theory[5] and variational perturbation theory[6,7] that the best choice of a variational parameter is the value at which the targeted observable is least sensitive to the change of the parameter. This technique is called the principle of minimal sensitivity (PMS). In refs. [7–10], it was shown that the perturbative expansion optimized with the PMS can succeed even when interaction is strong, and regular perturbation theory fails. In this work, we optimize the screening parameter $\lambda$ with PMS and observe a vast improvement to the convergence of the targeted response functions with expansion order.

**The sign-blessed Monte Carlo algorithm.** While our setup of the expansion (with the static screening and the physical Fermi surface) is not entirely new[11–14], its evaluation to high-enough order until ultimate convergence has not to our knowledge been achieved before in any realistic model containing the long-range Coulomb interactions that are relevant for realistic solids. Our solution employs a recently developed diagrammatic Monte Carlo method[15–22], which is here optimized to take a maximal advantage of the sign blessing in fermionic systems[16]. Namely, by carefully arranging and grouping the Feynman diagrams, it is possible to ensure a massive sign cancellation for different diagrams in the same group, before the MC sampling is performed[21,23]. The conventional diagrammatic Monte Carlo algorithms[15–20], which were sampling the diagrams one by one, are highly inefficient here.

We evaluate diagrams in the momentum and imaginary-time representation, and for each configuration of random momenta $(\mathbf{k}_0, \mathbf{k}_1, \mathbf{k}_2, \cdots, \mathbf{k}_N)$ and times $(\tau_1, \tau_2, \cdots, \tau_{2N})$ generated by the Markov chain, we sum the contribution of all diagrams at a given order $N$, which have the same number of momenta and time variables[23]. For example, when computing the polarization at order $N = 6$, the sector without counterterms contains 14,593 Feynman diagrams (see Fig. 1). These are regrouped into a much smaller number of sign-blessed groups to boost the efficiency of the MC sampling. For example, motivated by the crossing symmetry, at the lowest order in the crossing exchange, we get from standard Feyman diagrams to so-called Hugenholtz diagrams[24] where the direct and exchange interactions are combined into an antisymmetrized four point vertex (see Fig. 1, green box). That exchange operation keeps the diagram exactly the same, except for a change of the overall sign and a change of momentum on a single interaction line, hence the pairs of such diagrams largely cancel. After this operation, there are only 877 Hugenholtz diagrams at order 6. To further reduce the number of diagrams, we then combine the polarization diagrams that can be derived from the same free-energy diagram by attaching two external vertices to propagators. Mathematically, adding external vertices to a free-energy diagram corresponds to taking its functional derivative with respect to the inverse propagator. Therefore, the above step groups the polarization diagrams into a conserving group in the Baym–Kadanoff sense[25], and the sign cancellation is guaranteed by the Ward identities (see Supplementary Note 1). For example, at order $N = 6$ there are only 41 such free-energy groups (see Fig. 1). We thus managed to reduce the complexity from 14,593 individual diagrams to 41 groups. The diagrams in the same group are very similar, and hence can share the identical momentum/time variables (except the external vertices). This not only ensures the massive sign cancellation between different diagrams but also reduces the cost of computing the total weight of Feynman diagrams in Monte Carlo updates.

Finally, beyond variationally optimizing the zeroth order term $(L_0)$ we can also look for improvement of the high-order vertex functions. One of our choices is to sum up all ladder diagrams dressing the vertices (see Supplementary Fig. 3). We call this scheme the Vertex Corrected Constant Fermi Surface (VCCFS). The original diagrammatic expansion is here called Constant Fermi Surface (CFS) scheme. The name originates in the above described principle that electron potential $\nu_{\mathbf{k}}$ is determined in such a way that $L$ and $L_0$ share the same physical Fermi surface volume.

**Spin susceptibility of the uniform electron gas.** All results in this work are obtained at temperature $T = 0.04E_F$, which is much lower than any other scale in UEG problem[26]; hence, results are the zero temperature equivalent. We want to point out that finite temperature calculations are very hard in the Diffusion Monte Carlo (DMC)[27], while our method is very well suited for finite temperature calculations, and converges even faster with the increasing order. While wave function properties, such as energy

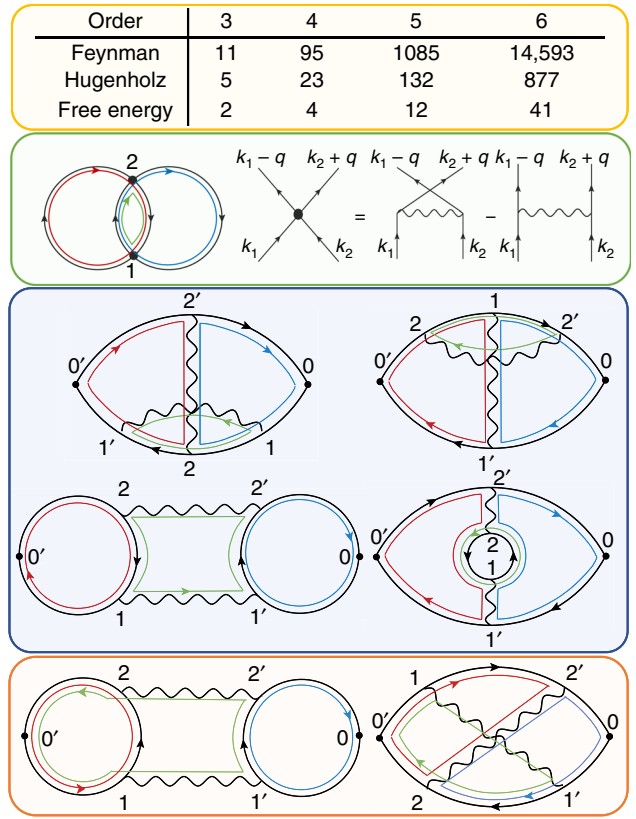

| Order | 3 | 4 | 5 | 6 |
|---|---|---|---|---|
| Feynman | 11 | 95 | 1085 | 14,593 |
| Hugenholz | 5 | 23 | 132 | 877 |
| Free energy | 2 | 4 | 12 | 41 |

**Fig. 1** Grouping of Feynman diagrams. The grouping is achieved by leveraging the fermionic crossing symmetry and the free-energy generating functional. Orange top box shows the number of Feynman/Hugenholtz/ Free-energy Hugenholtz diagrams at orders 3–6, excluding the Hartree–Fock sub-diagrams (see Supplementary Note 1). The green panel on the left and the right shows an example of the free-energy Hugenholtz diagram, and how is the Hugenholtz vertex related to the standard Feynman diagram. Note that a single Hugenholtz diagram with $N$ vertices (black dots) represents up to $2^N$ standard Feynman diagrams with alternating signs. By attaching two external vertices to different propagators in the Hugenholtz free-energy diagram in the green box, one generates four topologically distinct groups of standard Feynman diagrams for the polarization function. Two of them are shown in the blue and orange box below. By the process of attaching external vertices to a single Hugenholtz free-energy diagram, we generate 10 out of 11 standard Feynman diagrams for the polarization at the third order. The color lines represent our choice for momentum loops, which are uniquely determined by the choice of the loops in the free-energy Hugenholtz diagram. The external momentum is added through the shortest path connecting two external vertices. Note that such grouping of diagrams allows us to calculate the weight of all diagrams in this figure with only eight different electron propagators, instead of expected 36. The above protocol can generate multiple copies of the same Feynman diagram. We tested two approaches: (a) we keep one copy to avoid double counting; (b) we keep all copies and precompute the symmetry factors, and we notice somewhat better performance of scheme (b)

and pair distribution function, are very precisely computed by DMC[27], and some of them are also are amenable to approximations such as GW[28,29], the response functions are more challenging to evaluate with the existing techniques. The strength of our approach is that it can be used to compute both the static and the dynamic, the single and the multiparticle correlation functions.

In Figs. 2 and 3 we show the momentum-dependent (Pauli) spin susceptibility at zero frequency, which has never been

precisely calculated before to our knowledge even though its overall shape is crucial for the design of appropriate exchange-correlation functionals of the density functional (DFT) to predict magnetic order in real materials. In panels (a) and (b) we show how the convergence properties of the susceptibility $\chi_s$ depends on the screening parameter $\lambda$ in the theory. Note that the static screening in $L_0$ is always compensated by the counterterm in $\Delta L$, so that for any value of $\lambda$ the UEG model is recovered at infinite order limit. The observable $\chi_s(q = 0)$ develops a broad plateau as a function of $\lambda$ (Fig. 2a, b) at the point $\lambda_N^*$, which is slightly increasing with the increasing order. This shows that if expansion is carried out to high-enough order, the physics becomes more and more local and allows one to use very short-range form of the interaction, which greatly improves the efficiency of the method. We note that this property will be very beneficial in the realistic material applications, where the converged result is extremely difficult to obtain due to the long-range nature of the bare Coulomb interaction. Figure 2c shows the value of $\chi_s(q = 0)$ at the optimized $\lambda_N^*$ versus perturbation orders. When the PMS is used, such that the variational parameter $\lambda$ is optimized order by order, the convergence is very rapid, even when the bare interaction is strong. The value $\chi_s(q = 0)$ at the optimized $\lambda_N^*$ is monotonically increasing with the increasing order in the CFS scheme, and beyond the second order is oscillating around the converged value in VCCFS scheme. Both schemes converge towards the same value, and the systematic error bar at a given truncation order can be estimated from comparison between the two methods, allowing one to extract very precise value of $\chi_s(q = 0)$ even at a moderate expansion order (see Fig. 2c, Table 1).

Figure 2d shows the momentum dependence of spin susceptibility $\chi_s(q)$ at $\lambda^*/E_F = 0.75$, optimized at the highest order ($N = 6$) and its comparison to the non-interacting (RPA) result, which underestimates $\chi_s$ up to 57%.

In Fig. 3a we show the same spin susceptibility as in Fig. 2d, but for other values of density parameter $r_s = 1$–4 (here density $n = 3/(4\pi r_s^3)$). Both VCCFS and CFS schemes agree with each other within the statistical error bar at order $N = 6$ for all $r_s \leq 4$. We note that this spin susceptibility plays a central role in construction of the DFT exchange-correlation kernel for magnetically ordered systems. Finally, Fig. 3b displays the static local-field correction, which measures the deviation from the non-interacting electron gas ($\chi_{RPA}$), $G(q) \equiv \frac{q^2}{8\pi}(\chi_{RPA}^{-1}(q, \omega = 0) - \chi^{-1}(q, \omega = 0))$. It is a very sensitive measure of electron correlations. It has been suggested in the literature that the possible peak near $k \sim 2k_F$ is of great importance for understanding the quasiparticle properties[30]. Within the local density approximation, the function $G(q)$ is approximated by the quadratic parabola depicted in Fig. 3b[31], which is an excellent approximation at small $q \leq k_F$, but its deviation from the quadratic function is very pronounced near $2k_F$. Note that within RPA $G(q)$ vanishes, as RPA does not take into account the exchange-correlation kernel. We note that our calculation clearly shows that in the exact solution, the local-field correction displays non-trivial maximum just above $2k_F$, which has not been established before.

**Charge response function**. Figure 4 shows the dielectric function $\epsilon(q)$ for densities $r_s = 1$ to $r_s = 4$, and its comparison to RPA and DMC[31,32] results. We show several orders ($N = 2$–5) using VCCFS scheme, and also the extrapolated result to $N = \infty$ using standard second-order Richardson extrapolation. The DMC data are in agreement with our prediction, but notice that DMC allows one to calculate only a set of discrete points, while the developed VDMC method gives a smooth and very accurate continuous curve, which allows one to resolve the fine structure. For example, we notice that there is a clear kink of $1/\epsilon$ curve near $2k_F$. This

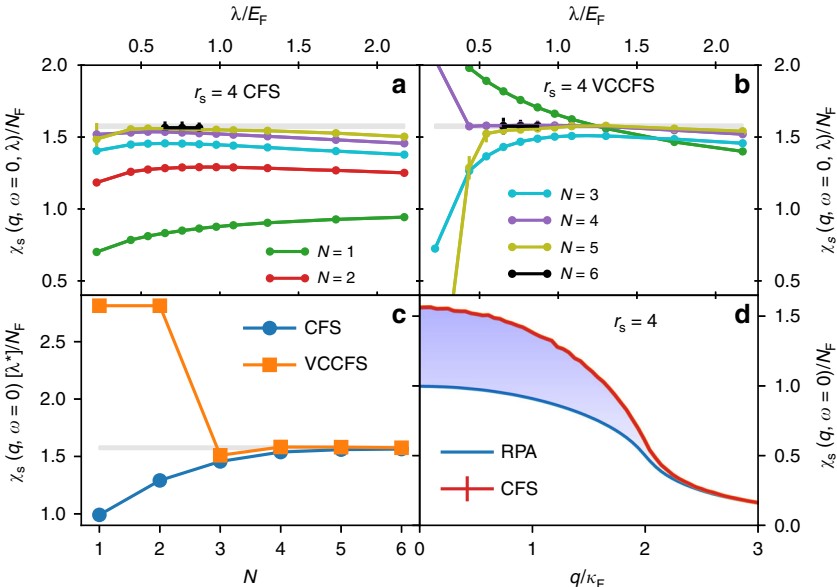

**Fig. 2** Spin susceptibility of UEG at $r_s = 4$. This corresponds to a density $n = 3/(4\pi r_s^3)$. The optimization of $\chi_s(q = 0, \omega = 0)$ versus the screening parameter $\lambda$ within **a** CFS and **b** VCCFS scheme. Susceptibility $\chi$ and $\lambda$ are scaled by the density of states at the Fermi level $N_F = (\frac{3}{2\pi})^{2/3}/(2\pi r_s)$, and the Fermi energy $E_F$, respectively. The shaded region shows the estimated total s.d. error bar of our calculation. A single extrememum at the optimized $\lambda^*$ appears, which is however order dependent ($\lambda_N^*$). **c** The value of the optimized $\chi(q = 0, \omega = 0)[\lambda_N^*]$ versus diagram order in both schemes. **d** The momentum-dependent $\chi(q, \omega = 0)$ at the converged order $N = 6$ and optimized $\lambda_{N=6}^*/E_F = 0.75$ in CFS scheme, along with comparison to random phase approximation (RPA), which is exact when interaction is ignored. The statistical s.d. errors are displayed in panels **a**, **b** and **d**, and in **d** are smaller than the width of the curve

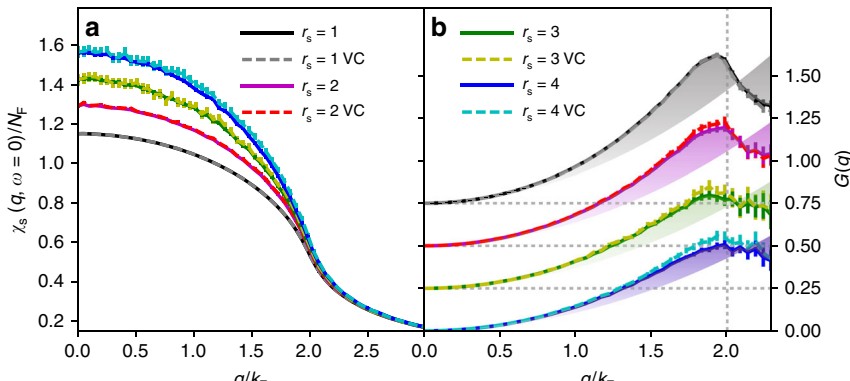

**Fig. 3** Spin susceptibility and static local-field correction. **a** $\chi_s(q, \omega = 0)$ at optimized $\lambda^*$ for $r_s = 1$–4. VC corresponds to VCCFS scheme, and the rest to CFS scheme. The statistical s.d. error bars are displayed for each computed point, and each point is computed statistically independently. In VCCFS scheme the statistical error bars are larger than in CFS scheme, but agree with each other within the error bar. **b** The local-field correction for the same $r_s = 1$–4, and its deviation from quadratic approximation (see the color envelope). For clarity the curves for $r_s = 1$, 2, and 3 are shifted up for 0.75, 0.5, and 0.25

**Table 1 Long wavelength values of spin and charge response**

| $r_s$ | $\chi_s/N_F$ | litt.($\chi_s/N_F$) | $P(0)/N_F$ | litt.($P(0)/N_F$) |
|---|---|---|---|---|
| 1 | 1.152 (2) | 1.15–1.16 | 1.208 (6) | 1.207–1.208 |
| 2 | 1.296 (6) | 1.27–1.31 | 1.54 (2) | 1.549–1.549 |
| 3 | 1.438 (9) | 1.39–1.48 | 2.20 (6) | 2.194–2.203 |
| 4 | 1.576 (9) | 1.51–1.66 | | |

First column $\chi_s = \chi_s(q = 0, \omega = 0)$ is the spin susceptibility, here normalized by the density of states at the Fermi level ($N_F$), as computed by the current method. The second column shows the range of previous estimations from the literature[37]. $P(0) \equiv P(q = 0, \omega = 0)$ is the static uniform charge polarization as obtained by this method. Unfortunately both CFS and VCCFS methods approach the converged value from below, hence extrapolation to $N = \infty$ is needed, which leads to much larger error bar in our calculation. The fourth column lists previous DMC results, extracted from two different correlation energy ansatzes proposed in refs. [37,38]

feature has been proposed in some theories (e.g. ref. [33]), but the previous DMC results in refs. [31,32] were not precise enough to confirm or disprove it.

Finally, in Table 1 we give our best estimates for the static spin and charge response with estimation of the error bar. Within our method the spin response shows faster convergence with increasing order; hence, it allows us to compute the spin response more precisely than the charge response. Therefore, our values for $\chi_s/N_F$ are more precise than currently available literature (compare columns one and two). Note that the previous estimate for the spin susceptibility relied on an uncontrolled ansatz for the spin dependence of the susceptibility, hence large uncertainty.

Contrary to the spin response, or finite momentum charge response, the static uniform charge response $P(q = 0, \omega = 0)$ can

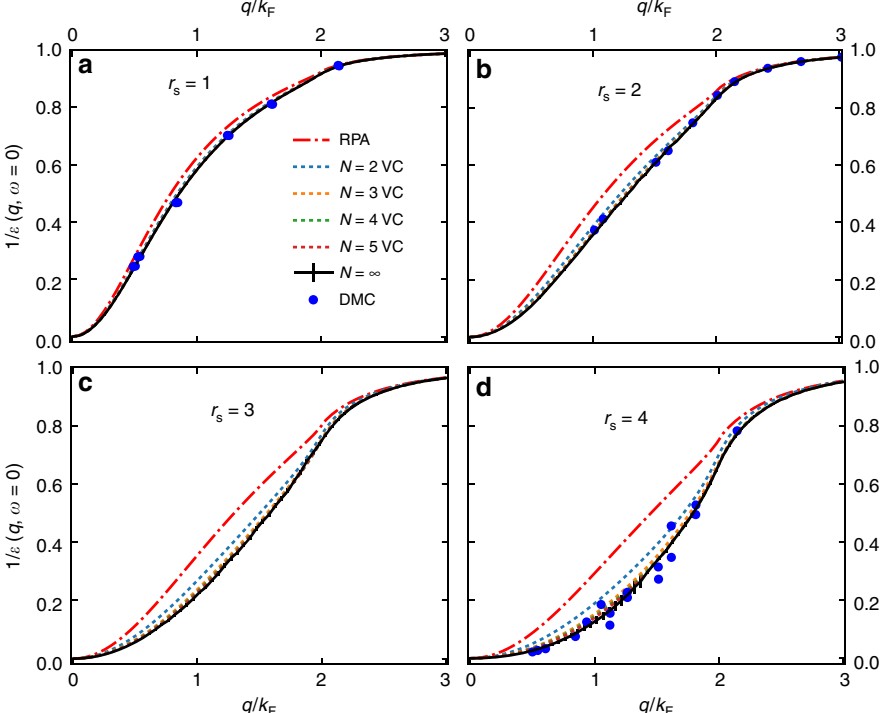

**Fig. 4** The inverse dielectric function ($1/\epsilon$). **a–d** Results for $r_s = 1$–4 respectively at $\lambda^*_{N=5}$, optimized for order 5, but we show $1/\epsilon$ for all orders up to 5 and its extrapolated value. We also display error bars for extrapolated curve, which contains both the statistical error and the estimated extrapolation error. Here we use more rapidly converging VCCFS scheme. The comparison to DMC and RPA is shown. The DMC data are from refs. [31,32]

be obtained from the ground state energy of the system, without explicitly introducing a modulated external potential, and hence it can be extracted very precisely from the existing DMC calculations. We compare it with our results, and find excellent agreement. We note that static $P(q = 0, \ \omega = 0)$ at $r_s = 4$ convergences very slowly in our method, due to proximity to the well known charge instability at $r_s \approx 5.2$; hence, we can not reliably extrapolate its value to infinite order at $r_s \geq 4$.

## Discussion

The prospects of combining the Variational diagrammatic Monte Carlo with DFT to obtain theoretically controlled results in real solids are particularly exciting, as the DFT potential is semi-local and can be added to $v_k$, so that it will play a role of a counter-term in the expansion. The complexity would be modest, because no expensive self-consistency is required, and because the interaction is statically screened, hence the scaling of this method should be similar to the complexity of screened hybrids[34] rather than the self-consistent GW approximation[35].

## Methods

**Model definition**. The UEG model describes electrons in a solid where the positive charges, which are the atomic nuclei, are assumed to be uniformly distributed in space. The electrons interact with the other charges through a long-range Coulomb interaction. The second-quantized Hamiltonian is

$$\hat{H} = \sum_{\mathbf{k}\sigma} \left(\mathbf{k}^2 - \mu\right) \hat{\psi}^\dagger_{\mathbf{k}\sigma} \hat{\psi}_{\mathbf{k}\sigma} + \tag{1}$$

$$\frac{1}{2V} \sum_{\substack{\mathbf{k}\mathbf{k}'\sigma\sigma' \\ \mathbf{q}\neq 0}} \frac{8\pi}{q^2} \hat{\psi}^\dagger_{\mathbf{k}+\mathbf{q}\sigma} \hat{\psi}^\dagger_{\mathbf{k}'-\mathbf{q}\sigma'} \hat{\psi}_{\mathbf{k}'\sigma'} \hat{\psi}_{\mathbf{k}\sigma} \tag{2}$$

where $\hat{\psi}/\hat{\psi}^\dagger$ are the annihilation/creation operator of an electron, $\mu$ is the chemical potential controlling the density of the electron in the system. We measure the energy in units of Rydbergs, and the wave number $\mathbf{k}$, $\mathbf{q}$ in units of inverse Bohr radius.

**Lagrangian with the counterterms**. In the path integral representation, using the standard Hubbard–Stratonovich transformation, the Lagrangian of the uniform electron gas can be cast into the form in which the Coulomb interaction is mediated by an auxiliary bosonic field $\phi_{\mathbf{q}}$. Motivated by the well known fact that the long-range Coulomb interaction is screened in the solid, and that the effective potential of emerging quasiparticles differs from the bare potential, we introduce the screening parameter $\lambda_{\mathbf{q}}$ and an electron potential $v_{\mathbf{k}}$ into $L_0$, which then takes the form

$$\begin{aligned} L_0 &= \sum_{\mathbf{k}\sigma} \psi^\dagger_{\mathbf{k}\sigma} \left(\frac{\partial}{\partial\tau} - \mu + \mathbf{k}^2 + v_{\mathbf{k}}(\xi=1)\right) \psi_{\mathbf{k}\sigma} \\ &\quad + \sum_{\mathbf{q}\neq 0} \phi_{-\mathbf{q}} \frac{q^2 + \lambda_{\mathbf{q}}}{8\pi} \phi_{\mathbf{q}} \end{aligned} \tag{3}$$

and represents well the low-energy degrees of freedom in the problem when parameters $\lambda_{\mathbf{q}}$ and $v_{\mathbf{k}}$ are properly optimized. To compensate for this choice of $L_0$, we have to add the following interaction:

$$\Delta L = -\sum_{\mathbf{k}\sigma} \psi^\dagger_{\mathbf{k}\sigma} v_{\mathbf{k}}(\xi) \psi_{\mathbf{k}\sigma} - \xi \sum_{\mathbf{q}\neq 0} \phi_{-\mathbf{q}} \frac{\lambda_{\mathbf{q}}}{8\pi} \phi_{\mathbf{q}} \tag{4}$$

$$+ \sqrt{\xi} \frac{i}{\sqrt{2V}} \sum_{\mathbf{q}\neq 0} \left(\phi_{\mathbf{q}} \rho_{-\mathbf{q}} + \rho_{\mathbf{q}} \phi_{-\mathbf{q}}\right). \tag{5}$$

so that, when the number $\xi$ is set to unity, $L = L_0 + \Delta L(\xi)$ is exactly the UEG Lagrangian. The density $\rho$ is $\rho_{\mathbf{q}} = \sum_{\mathbf{k}\sigma} \psi^\dagger_{\mathbf{k}\sigma} \psi_{\mathbf{k}+\mathbf{q}\sigma}$. Note that the first two terms in $\Delta L$ are the counterterms[14] which exactly cancel the two terms we added to $L_0$ above. We use the number $\xi$ to track the order of the Feynman diagrams, so that order $N$ contribution sums up all diagrams carrying the factor $\xi^N$. We set $\xi = 1$ at the end of the calculation. Note also that this arrangement bears similarity with the well established methods, such as G0W0[35], which computes the self-energy at the lowest order ($\xi^1$) and sets $v_{\mathbf{k}}$ to the DFT Kohn–Sham potential, and $\lambda_{\mathbf{q}}$ to the bubble diagram ($\lambda_{\mathbf{q}} = g^0 g^0$ with $g_{\mathbf{k}}^{0-1} = (i\omega + \mu - \frac{\mathbf{k}^2}{2m} - v_{\mathbf{k}})$). The so-called skeleton Feynman diagram technique is recovered when $v_{\mathbf{k}}$ and $\lambda_{\mathbf{q}}$ are equated with the self-consistently determined self-energy and polarization. However, note that such diagram expansion can be dangerous, as it can lead to false convergence to the wrong solution[36].

**Variational optimization**. In optimizing the screening parameter $\lambda_{\mathbf{q}}$ by the principle of minimal sensitivity, we found it is sufficient to take a constant $\lambda_{\mathbf{q}} = \lambda$. Furthermore, we found that the uniform convergence for all momenta is best

achieved when the electron potential $v_{\mathbf{k}}$ preserves the Fermi surface volume of $g_{\mathbf{k}}^0$; therefore, we expand $v_{\mathbf{k}} = \xi(\Sigma_{\mathbf{k}}^{\mathrm{x}} - \Sigma_{\mathbf{k}_F}^{\mathrm{x}}) + \xi^2 s_2 + \xi^3 s_3 \cdots$, and we determine $s_N$ so that all contributions at order $N$ do not alter the physical volume of the Fermi surface. In other words, we ensure the density, which can be calculated with the identity $n = -P_{\mathbf{q}}(\tau = 0)$ where $|\mathbf{q}| \gg k_F$, remains fixed order by order. Since the exchange ($\Sigma_{\mathbf{k}}^{\mathrm{x}}$) is static, and is typically large, we accomodate it at the first order into the effective potential, so that at the first order we recover the screened Hartree–Fock approximation, i.e., interaction screened to $\sim \exp(-r\sqrt{\lambda})/r$ and optimized $\lambda$.

We also introduce a vertex correction scheme (VCCFS) to further improve the convergence of the series. In practice, within the VCCFS scheme, we precompute the three-point ladder vertex, and attach it to both sides of a polarization Feynman diagram, and at the same time, we eliminate all ladder-type diagrams from the sampling, to avoid double counting of diagrams (see Supplementary Fig. 4).

**Comparison to existing approaches**. Finally, we discuss the advantages and limitations of the proposed method. The current variational approach is very effective at weak to intermediate correlation strength (spin/charge response up to $r_s \approx 4$), but to extend it to the regime with stronger correlations, one would need to introduce more sophisticated counterterms, such as the three and the four point vertex renormalization, to capture the emergent charge instability around $r_s \approx 5.2$. Beyond the variational approach, we also want to point out that our developed Monte Carlo algorithm is a very generic Feynman diagram calculator for many-electron systems with long-range Coulomb repulsion, and is more efficient and simpler that the existing conventional diagrammatic Monte Carlo of refs. [15–20]. For example, VDMC requires only three updates, while the conventional approach needs about dozen updates. More importantly, this algorithm utilizes the sign-blessed grouping techinque to dramatically improve the sampling efficiency. Comparing to the recently proposed Determinant Diagrammatic Monte Carlo algorithm[21], our method is more generic in the sense that the algorithm can directly work in any representation (momentum/frequency, space/time) and can handle any vertex renormalization without sacrificing the efficiency.

## Data availability
All analytical data not given in the Supplementary Information are available on request.

## Code availability
The code is available at https://github.com/haulek/VDMC with https://doi.org/10.5281/zenodo.3309567, and https://github.com/kunyuan/FeynCalc with https://doi.org/10.5281/zenodo.3308233.

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

## Acknowledgements
We thank G. Kotliar, N. Prokof'ev, B. Svistunov, and Y. Deng for stimulating discussion. This work is supported by the Simons Collaboration on the Many Electron Problem and NSF DMR-1709229.

## Author contributions
Both K.C. and K.H. developed the MC code, created the theoretical formalism, carried out the calculation, and analyzed the results and wrote the paper. K.H. supervised the project.

## Additional information

**Competing interests:** The authors declare no competing interests.

