## [Peer Review File · Nature Communications]

Reviewers' comments:

Reviewer #1 (Remarks to the Author):

In their paper "Feynman's solution of the quintessential problem in solid state physics" the authors introduce a variational diagrammatic Monte Carlo approach to calculate response functions for the homogeneous electron gas in three dimensions. Their method makes it possible to do calculations which were unattainable before because of the sign problem. Having a technique for calculating the properties of the electron gas in a controlled way is of fundamental importance and opens the door to solving important problems in solid-state physics.

Many of the elements of the proposed technique are known in the literature, but it is their combination that makes this paper impressive. Key elements to make the technique work are the introduction of a Lagrangian containing a screened interaction in the solvable part, choosing the screening parameter in an order-dependent way according to the PMS such that convergence can be achieved, assigning the momenta of the Feynman diagrams in such a way that cancellation is optimized.

In my opinion, the paper merits publication in Nature Communications after some revisions and corrections. The paper contains a number of errors and a few points should be clarified.

- The diagrammatic Monte Carlo method evaluates and sums all Feynman diagrams at each order up to a certain expansion order. The authors write that the diagrams are regrouped into a much smaller number of "sign-blessed" groups. However, the weight of a 'configuration' is just the sum of the weights of *all* diagrams for a given set of internal variables. Therefore the division into groups seems irrelevant at first (since they are summed anyway). The only purpose of the groups seems to lie in the fact that they allow to assign the internal variables in an optimal way (to enhance cancellation). This point seems to be somewhat hidden on page 2 of the paper. At first, the explanation gave the impression that the number of sampled diagrams was reduced (by sampling Hugenholtz diagrams). I think the paper would be much more clear if this point is made more clearly.

- Figure 1 is very useful, but raises a number of unanswered questions. The shown free energy Hugenholtz diagram gives rise to 2^2 diagrams after splitting the interactions. By cutting 2 of the 4 propagators, one gets thus $2^2 * 6$ possible polarization diagrams. I was thus expecting 6 groups of 4 diagrams (with possibly double counting or reducible diagrams which can be removed). Group 1 shows indeed 4 diagrams, but group 2 has only two diagrams. It would be nice to explain why. Moreover, it is stated that there are only 4 groups. The remaining diagrams are dropped because they are reducible / already included / Hartree-Fock? Please explain. If the procedure produces the same topology multiple times, how does one decide which diagrams to keep (they might still differ by the choice of internal momenta, which affects the sign cancellation)? If one simply cuts a line, then, due to momentum conservation, the external momentum must be zero. How do you make the external momentum of the polarization finite? Do you add the momentum along some path connecting the two external vertices?

- There are clearly some problems with the Lagrangian $L_0 + \Delta L$. The last term in Eq. (1) is proportional to e^2 , but the e has also been included in the coupling between bosonic fields and fermionic fields (last term in Eq. (2)). Therefore each order gets a e^4 . The prefactor in the last term of Eq. (1) should not be there, to make the notation consistent with $8 \pi / (q^2 + \lambda_q)$ for the bosonic propagator (as is written in figure 1 of the supplemental material). In Eq. (2), the last two terms which couple bosons to fermions should be conjugate. They are clearly not. I would like to ask the authors to fix all the problems with this Lagrangian and to arrive at a consistent notation (which is also consistent with the notation in figure 1 of the supplementary material).

- It is not clear to me why the bare Coulomb interaction in momentum representation is taken to be $8 \pi e^2 / q^2$ rather than $4 \pi e^2 / q^2$. Do the authors take some factor 2 into

account because of spin? Please explain.

- On page 3, there is written "...so that with increasing expansion order the susceptibility converges towards the exact result." and "When the MPS is used... the convergence to the exact results is very rapid.." I think it is misleading and dangerous to write this. The original series for fixed λ is asymptotic (because subject to Dyson's collapse argument) but probably Borel resummable. Amazingly, the PMS allows to convert this divergent series into a convergent one. It has been rigorously shown to converge to the exact answer for the anharmonic quantum oscillator. For the current problem, however, I do not know of any theorem which guarantees that the scheme will converge to the exact physical answer. It would be more fair if the authors acknowledge this as an open problem (or add a reference to a proof of convergence).

- The insets in Figure 4b and 4d show the polarization as function of $1/N$ for fixed λ . Why not for λ^*_N ? Is this because λ^*_N is essentially independent from N for these values of N ? Note that in the figure caption you use the notation λ^* whereas in the figure you use just λ . Please correct this.

- In the table, no value is given for the polarization for $r_s=4$. Is this because the extrapolation to $N=\infty$ becomes simply unreliable? It is stated that no extrapolation for the spin response is needed since CFS and VCCFS approach from below and above, and that therefore the estimate of the spin response is more precise. But looking at the figures, it seems that the spin response just converges faster and is therefore more precise than the charge response. Is this correct? A related more general question: is the calculation limited to $r_s=1-3$? I think it is important to say something about the range in r_s .

- How do the authors estimate the final error bars? I think it is important to mention this since the whole point of the paper is to make quantitative and unbiased predictions.

- The expansion of v_k is taken such that the lowest order in ξ gives the Fock self-energy Σ^x minus $\Sigma^x_{\{k_F\}}$. The higher order terms are just shifts in the chemical potential to ensure that the density remains fixed at each order of the diagrammatic expansion. How are these shifts s_N determined in practice? It seems that this requires calculating the density. Because of this choice, Fock diagrams should be excluded. However, there is a Fock diagram at the second order in ξ in Figure 2 of the supplementary material.

Minor points:

In the caption of figure 2, $n = 2 / (4 \pi r_s^3)$ instead of $n = 3 / (4 \pi r_s^3)$. The Bohr radius is taken to be one here. Please write this somewhere.

In figure 2 and its caption: the authors introduce a λ^* and a λ^*_N for one and the same thing. It would be much more clear to keep just λ^*_N everywhere.

It sounds a bit bizarre to say that RPA is exact for the non-interacting electron gas, since it gives a screened interaction. Maybe it would be better to rephrase this.

Reviewer #2 (Remarks to the Author):

In the manuscript "Feynmann's solution of the quintessential problem in solid state physics", the authors present a novel diagrammatic Monte Carlo (DiagMC) scheme and apply it to the uniform electron gas (UEG). The UEG has long been used as a minimal model of, e.g., electrons in a metal and is of particular importance for the construction of exchange-correlation functionals for density functional theory.

While Diffusion Monte Carlo (DMC) has been very successfully applied to the UEG and has arguably remained the state-of-the-art method for this model since the seminal work by Ceperley and Alder [11], bias introduced by the fixed-node approximation, the need for finite-size extrapolation, and challenges in addressing finite temperature properties or response functions make new accurate and controlled methods highly desirable. The present manuscript presents such a new approach and at least matches the accuracy of DMC predictions for the charge and spin response functions, where available, while not being subject to fixed-node, finite-size, or zero-temperature limitations.

DiagMC denotes a rather recent computational approach to quantum many-body systems based on the stochastic sampling of Feynman diagrams. During the last ten years, it has been applied rather successfully to a wide range of interacting quantum systems, including the unitary Fermi gas, unconventional superconductors, and other correlated lattice models. Fermions with long-range interactions in the continuum, however, which host a certain amount of complications due to short- and long-wavelength divergences, have not been adequately addressed before.

In general, the two fundamental challenges in applying the DiagMC technique are (a) does the diagrammatic series converge sufficiently well to allow for an extrapolation of the calculated finite-order results to the infinite-order limit; and (b) how high a diagram order can be calculated with sufficiently small stochastic errors before the sign problem becomes too severe. The present work makes major improvements on both fronts.

First, the diagrammatic expansion's starting point is optimized to be close to the emergent Fermi liquid physics of the result. The basic idea of shifting the starting point with an artificial potential has been employed before [Ref. 23 and references therein], but the automatic optimization of these free parameters via the principle of minimal sensitivity and by fixing the Fermi surface volume have not been proposed in this context before, to my knowledge. Second, by grouping similar diagrams into classes whose constituents largely cancel each other, the present work achieves a marked alleviation of the sign problem for given diagram order. Other recent DiagMC works [cf. Ref. 25; also <https://doi.org/10.1103/PhysRevB.97.085117>, <https://arxiv.org/abs/1712.10001>] have explored determinantal schemes that achieve a similar effect by summing all diagrams at a given order, although the computational scheme is quite different in practice.

In total, this work presents a major achievement, which brings the DiagMC technique into the realm of ab-initio simulations for materials. I hence recommend its publication. However, additional details on the method should be provided (possibly in the supplement) and the presentation may be improved in some places, as detailed below.

- The actual, stochastic, DiagMC sampling process has not been described at all and it is not obvious that the conventional DiagMC procedure as described, e.g. in [Ref. 6], can be applied straightforwardly to the current setup. For example, Hugenholtz and free energy diagrams come with combinatorial prefactors, which are not easily determined in a local update. Also, given the moderate number of free energy diagrams up to sixth order (Fig. 1), one may plausibly perform the sum over topologies deterministically. Hence my request to specify what is actually done. Similarly, for the optimization of λ please specify what quantity is minimized (what susceptibility, at what 4-momentum). A dedicated overview of the computational procedure may also give readers less familiar with DiagMC a better impression of what your algorithm encompasses.

- Stochastic error bars should be quantified, and displayed in the figures.

- According to the authors, their CFS and VCCFS schemes are conserving approximations in the Baym-Kadanoff sense. Wouldn't this require a self-consistent definition of the diagrams in terms of

the interacting propagator?

- I am not sure I follow the authors in their discussion of Fig. 4: "The latter [DMC] data are in agreement with our prediction, although the systematic error-bar at any finite q , which are not given in the original papers but can be inferred from the fluctuations of the data points around our smooth curves, is much larger than ours, demonstrating the superiority of the newly developed VDMC method." (That conclusion is also echoed in the abstract.)

To some extent, my lack of persuasion may be an issue of the figure, which is rather overloaded and differences between curves are of similar size as line widths and symbol sizes. Furthermore, it is unfortunate that the DMC references do not give error estimates, and the present authors should not repeat the same mistake by not giving error bars for their own data. Lastly, the compared DMC work is over twenty years old, it is fair to assume that they would do better with modern computers and codes.

Hence, if the authors do want to show superiority of their method in this particular measure, they should plot data on a scale where differences are meaningful, and provide clear evidence that their errors are significantly smaller than what could be produced with DMC. Otherwise, the conclusions from the comparison should be restricted to consistency with DMC. This does not detract from the new method's additional capabilities compared to DMC, as mentioned above, and the value of being able to check predictions with two completely independent methods (fixed-node, finite-size vs. finite-order, thermodynamic limit).

Further suggestions for the authors' consideration:

- The "DVCCFS" scheme is rarely mentioned and does not seem to add much to the manuscript. Cutting it from the main text and figures may streamline the presentation.

- The "VCCFS" scheme seems to be sometimes called "LVCCFS". This inconsistency should be fixed.

- In the last paragraph of p. 2, the reference should be to Fig. 3, rather than 1, of the supplement.

- In Figs. 2(a, b) and 4(a), it would make sense to indicate the position of λ_N^* on each curve.

- "Hartree" is misspelled "Hatree" a couple of times.

- Fig. 3(b): Systematic and/or stochastic errors apparently become a lot more pronounced at large momenta. I wonder if the authors have any insight why that is the case?

- Fig. 4: As hinted above already, this figure is completely overloaded and it is plainly impossible to discern all the different curves. I would suggest to separate convergence plots - observable vs order at selected k points - from plots showing the k -dependence - extrapolated observable (only, or possibly + one finite-order result) with error bars vs k .

- Below Fig. 4: The V_q appearing in the definition of the polarization function is not defined anywhere, as far as I can see.

- It would be helpful to state a concise definition of the UEG model somewhere in the main text or methods section, say, by stating the Hamiltonian and units. While specialists can certainly defer these from the Lagrangian, it is unnecessarily complicated by the HS transform and variational parameters.

- I am afraid the use of the term "sign blessing" in the present work, while admittedly catchy, may further compound the existing confusion around this expression. To my knowledge, the term was

coined by Prokof'ev and Svistunov to point out that the fermionic sign can make a diagrammatic series convergent due to cancellation between terms with different sign at high orders. In that sense, "sign blessing" refers to a property of the series and is largely orthogonal to the sign problem. The "sign-blessed" diagram groups in the present context, in contrast, leave the series and its convergence properties unchanged but alleviate the sign problem by reducing the variance of the sampled distribution.

Reviewer #3 (Remarks to the Author):

The manuscript of Chen and Haule presents a new diagrammatic Monte Carlo method, building considerably on the work of a number of researchers in the last few years (of which I am not one). The work presents some impressive and convincing results in the case of the uniform electron gas in the thermodynamic limit – a significant development from previous (lattice) results, with the inclusion of a true long-range Coulomb interaction, and the computational of important two-particle quantities. The nice features of the approach include:

- Screened interaction in the zeroth-order terms, with optimized screening length
- Grouping of diagrams to control sign problem (though I strongly dislike the term 'sign-blessed'!)
- Two approaches to combine diagrams summed to infinity analytically
- Continuous momentum / imaginary time resolution
- Significant deviation from RPA-like behaviour found, even though there is essentially broad agreement with DMC.

The results are impressive in the accuracy they achieve for this paradigmatic model, and the manuscript is certainly worthy of careful consideration in this journal due to its important contribution to the field, and opening of a new avenue within this field. However, I have a number of comments which should be addressed before publication – some of a more general type, and others of a specific technical nature.

- The scope, drawbacks and limitations of the approach are not discussed (and therefore solely defined by what is not in the manuscript). More consideration should be given to the areas in which the method can particularly excel, and why. For instance, it is known that currently for lattice models, diagrammatic Monte Carlo is only predictive up to $U \sim 4t$ – presumably this carries over to r_s . This is despite the high- r_s limit (Wigner crystal) in some sense being easier, with the effective interaction shorter ranged. Presumably the limitation is that the convergence with N becomes slower as r_s increases? What is the limitation in going to higher r_s ? How does the explicit inclusion of a screened interaction in L_0 improve this? Furthermore, some discussion should be taken by the scaling of the computational effort both with N and r_s .

- It is well known certain diagrammatic expansions are only conditionally convergent, and that along with the 'pillars' being the quality of the starting point, and the ability to calculate contributions to high enough order, the other key question is the uniqueness of the result, and the assurance that the summation of diagrams will indeed converge via finite summation. What assurances are there of this for the diagrams which are summed, and the computational algorithm employed here?

- The first paragraph talks about the method being able to calculate the 'momentum-frequency' resolved response functions. However, I see no frequency resolution in any of the results presented, as all the results are simply giving the static limit. Why is no frequency-dependence given, as its omission looks like a limitation of the method. I am confused as to how the $w=0$ results were obtained even. The diagrams are sampled in τ -space, therefore presumably you are continuing the result to real-frequency (or integrating to get the static limit)? Is the lack of frequency resolution down to the uncertainty in the continuation, or the some other limitation which I am missing? All the results obtained could instead be computed just by looking at the results of static perturbations (as was done for the DMC comparisons). This is an important point which must be addressed, to determine the viability of the method for the true dynamics of the

system.

- Continuing on from the point above, surely the λ^* values would depend on both the momentum and frequency point considered. This would be especially true of changing frequency point, as (I presume) you would expect the λ^* value to tend to 0 in the high-frequency limit? If the aim is to get the full dynamics in one calculation, then this would not be possible?

- Furthermore, the optimization of λ is performed with a 'minimal sensitivity' principle. In effect, this is looking for a number of derivatives minimized in the variation of λ wrt the observable. However, it is seen that this is far from clear for lower values of N in figure 2, where there is a strong N dependence, and arguably for $N=1,2,3$, the least sensitive value of λ is at larger values than even plotted. Can the authors comment on this, and the sensitivity of λ^* wrt N (as well as momentum and frequency as discussed above).

- It is implied that for χ_s that the LVCFS and CFS bracket the $N \rightarrow \infty$ result for any finite N . This is presumably not proved, and while the numerical results seem to indicate this, it is not a rigorous result which can be relied on (as seen for the charge response).

- Similarly, there is presumably also no guarantee of monotonicity in the results, despite the groupings of diagrams and numerical evidence (this also somewhat relates to the formal convergence of the approach).

- Can something be explicitly said about whether all independent time-orderings of a particular diagram are included in a group, or whether these are considered in distinct groups – I was unclear on this from the manuscript.

- I was very confused at parts by the acronyms. DVCCFS is defined, but never seemingly used, while later on in the manuscript, LVCFS crops up, but is never defined. Is this latter acronym meant to be VCCFS? If DVCCFS is not used, then it should be removed. Furthermore, above Table I, a 'VDMC' acronym is used, and not defined?! Presumably this is meant to be LVCFS??

- I am confused by the assertion that 'Contrary to the finite momentum response, the static charge response $P(q=0, w=0)$ can be obtained from the ground state energy of the system'. Surely this is not true, and the momentum-resolution can also be obtained (see for instance the reference 31), where the momentum resolution is obtained. Furthermore, momentum resolved DMC results are given in Fig 4 anyway?!

- Figure 4 for has too many plots and legends to work out what is going on in a reasonable way, with the detail too small to be presentable. I am sympathetic to space constraints, but this is too compact.

- 'Feynman' is incorrectly spelt in the title of the manuscript (!)

- line 5: missing 'a'

- ' P_q with the increasing order The inset of Fig. 4b' (missing full stop)

Reply to Referee A

Referee: In their paper “Feynman’s solution of the quintessential problem in solid state physics” the authors introduce a variational diagrammatic Monte Carlo approach to calculate response functions for the homogeneous electron gas in three dimensions. Their method makes it possible to do calculations which were unattainable before because of the sign problem. Having a technique for calculating the properties of the electron gas in a controlled way is of fundamental importance and opens the door to solving important problems in solid-state physics.

Many of the elements of the proposed technique are known in the literature, but it is their combination that makes this paper impressive. Key elements to make the technique work are the introduction of a Lagrangian containing a screened interaction in the solvable part, choosing the screening parameter in an order-dependent way according to the PMS such that convergence can be achieved, assigning the momenta of the Feynman diagrams in such a way that cancellation is optimized.

In my opinion, the paper merits publication in Nature Communications after some revisions and corrections. The paper contains a number of errors and a few points should be clarified.

Reply: We thank the Referee for positive evaluation of our work.

Referee: The diagrammatic Monte Carlo method evaluates and sums all Feynman diagrams at each order up to a certain expansion order. The authors write that the diagrams are regrouped into a much smaller number of “sign-blessed” groups. However, the weight of a ‘configuration’ is just the sum of the weights of **all** diagrams for a given set of internal variables. Therefore the division into groups seems irrelevant at first (since they are summed anyway). The only purpose of the groups seems to lie in the fact that they allow to assign the internal variables in an optimal way (to enhance cancellation). This point seems to be somewhat hidden on page 2 of the paper. At first, the explanation gave the impression that the number of sampled diagrams was reduced (by sampling Hugenholtz diagrams). I think the paper would be much more clear if this point is made more clearly.

Reply: We thank the Referee for the suggestions. We have clarified the benefits of grouping diagrams in our manuscript on page 2, and also in the supplementary information. There are two purposes: 1) To achieve massive sign cancellation in Monte Carlo simulations, all diagrams in the same group should share the same basis for both momentum and time variables. This rule allows one to generate optimized basis for all the diagrams in the same group. 2) Since all diagrams in the same group share the same basis, they share most of the propagators and interactions. This allows fast evaluations of the total diagram weights for a given group, and only a small part of the diagram needs to be modified when evaluating the diagrams in the same group. For example, when evaluating diagrams in the same Hugenholtz group, we just need to make a very local change to sum two diagrams together, namely, change the value of the

interaction on a single bosonic propagator, while **all** electron propagators are unchanged, and all other interactions are unchanged. Thus on each interaction line we can sum two possible values for the interaction (with cross and without cross), and this allows us to sum up to 2^n propagators by making only n summations of interactions.

The number of samples diagrams is actually reduced, as one can basically sample Hugenholtz diagrams (with a small caveat explained below). The trick is that all Feynman diagrams, which correspond to the same Hugenholtz diagram (there are 2^N such diagrams), contain exactly the same fermionic propagators, hence we do not need to compute the same propagators and their product 2^N times, but only once. Moreover, the interaction lines are not identical, however, they contain a lot of common products. One can show that a binary tree can be constructed, with the depth equal to the number of Hugenholtz interaction propagaors, in which each vertex adds either the direct or the exchange interaction to the Hugenholtz diagram. The leaves of such a binary tree contain exactly 2^N terms, corresponding to the products we need to evaluate the sum of 2^N Feynman diagrams, while the number of operations to evaluate such a tree grows as $O(N)$. This is now also explained in the supplementary information.

[DONE]Referee: Figure 1 is very useful, but raises a number of unanswered questions. The shown free energy Hugenholz diagram gives rise to 2^2 diagrams after splitting the interactions. By cutting 2 of the 4 propagators, one gets thus $2^2 * 6$ possible polarization diagrams. I was thus expecting 6 groups of 4 diagrams (with possibly double counting or reducible diagrams which can be removed).

1) Group 1 shows indeed 4 diagrams, but group 2 has only two diagrams. It would be nice to explain why.

2) Moreover, it is stated that there are only 4 groups. The remaining diagrams are dropped because they are reducible / already included / Hartree-Fock ? Please explain.

3) If the procedure produces the same topology multiple times, how does one decide which diagrams to keep (they might still differ by the choice of internal momenta, which affects the sign cancellation)?

4) If one simply cuts a line, then, due to momentum conservation, the external momentum must be zero. How do you make the external momentum of the polarization finite? Do you add the momentum along some path connecting the two external vertices?

Reply:

Question (1),(2),(3) are similar, hence we answer them together. The protocol we proposed can generate multiple copies of the same Feynman diagram. Different copies have the same topology but different momentum/time basis. As noted in Question (1), the Hugenholtz exchange of propagators for group 2 generates four Feynman diagrams, but only two of them are topologically independent (the two other are redundant, but can be kept if desired). And as noted by referee in Question (2), by adding two external vertices to the generating functional in the top diagram of Fig.1, we generate six Hugenholz diagrams for the polarization. However, only four of them are topologically independent (again the two remaining are equivalent to diagrams that already appeared). We tested two approaches: a) we just keep one copy to avoid double counting. b) we keep all copies and precompute the symmetry factors. Both approaches are working and give similar results, however variant b) is slightly more efficient (gives 50% smaller

statistical error while it is 10% more expensive). However, even in variant a) the sign cancellation between diagrams is still strong enough for high order simulations.

As for Question (4), we add the external momentum along an arbitrary path between the two external vertices; however, in our experience it is best to choose the shortest path connecting the two external vertices. Note that this choice does not affect the sign cancellation in the small external momentum limit, where the sign problem is most severe.

We have now clarified the above issues in the manuscript (the caption of Fig.1) and in the supplementary material.

[DONE]Referee: There are clearly some problems with the Lagrangian $L_0 + \Delta L$. The last term in Eq. (1) is proportional to e^2 , but the e has also been included in the coupling between bosonic fields and fermionic fields (last term in Eq. (2)). Therefore each order gets a e^4 . The prefactor in the last term of Eq. (1) should not be there, to make the notation consistent with $8\pi / (q^2 + \lambda_q)$ for the bosonic propagator (as is written in figure 1 of the supplemental material). In Eq. (2), the last two terms which couple bosons to fermions should be conjugate. They are clearly not. I would like to ask the authors to fix all the problems with this Lagrangian and to arrive at a consistent notation (which is also consistent with the notation in figure 1 of the supplementary material).

Reply: We thank the Referee for pointing out inconsistency in the Eqs. (1) and (2). We used somewhat unusual notation in which “e” was not present in the interaction of the Hamiltonian (interaction had a form $8\pi/q^2$ because of atomic Rydberg units), while “e” was introduced in Hubbard Stratonovich transformation to count the order of the diagrams.

In the new version of the manuscript we start with the Hamiltonian of the electron gas in atomic units, and we derive the proper Hubbard Stratonovich Lagrangian without introducing (a redundant) charge “e”. Note that now we use Φ_{-q} and Φ_q so that $\Phi_{-q} = \Phi_q^*$, so that explicit conjugation is not needed. We are now using the notation, which is consistent with the book of Nagaosa (Quantum Field Theory in Condensed Matter Physics), except that we consistently use atomic units, while he is using Gaussian units. Note that we wanted to make quadratic part of the bosonic Lagrangian positive, therefore we added extra “i” to the boson-fermion coupling constant. One can redefine Φ to make it purely imaginary, which makes the coupling Hermitian, but the quadratic bosonic term negative. Both choices lead to the same Feynman diagrams.

[DONE]Referee: It is not clear to me why the bare Coulomb interaction in momentum representation is taken to be $8\pi e^2 / q^2$ rather than $4\pi e^2 / q^2$. Do the authors take some factor 2 into account because of spin? Please explain.

Reply: The bare Coulomb interaction is $e^2/(\epsilon_0 * q^2)$, which in units of Rydbergs becomes $8\pi/(q^2)$. Note that in Hartree units would be $4\pi/(q^2)$.

[DONE] Referee: On page 3, there is written “..so that with increasing expansion order the susceptibility converges towards the exact result.” and “When the MPS is used... the

convergence to the exact results is very rapid.. “ I think it is misleading and dangerous to write this. The original series for fixed λ is asymptotic (because subject to Dyson’s collapse argument) but probably Borel resummable. Amazingly, the PMS allows to convert this divergent series into a convergent one. It has been rigorously shown to converge to the exact answer for the anharmonic quantum oscillator. For the current problem, however, I do not know of any theorem which guarantees that the scheme will converge to the exact physical answer. It would be more fair if the authors acknowledge this as an open problem (or add a reference to a proof of convergence).

Reply: We thank the Referee for his/her suggestion. The convergence properties of the response functions in this work is based on numerical evidence, rather than rigorous mathematical proof. We believe that for large enough λ , the finite series is asymptotically converging down to a value exponentially close to the exact result, but eventually at even larger orders, the series will likely start to diverge. Because of our choice of the Lagrangian L_0 , the series is effectively weakly coupled, and similar to QED. In the QED it is possible to compute the “g” factor to very high accuracy (13 digits) at a finite order perturbation expansion, even though the series is not strictly converging, and it diverges at large enough perturbation order. However, we do not have a mathematical proof of such behavior, therefore we removed word “converges to the exact result” from the manuscript and pointed out that our conclusions of convergence is based only on numerical observation.

[DONE]Referee: The insets in Figure 4b and 4d show the polarization as function of $1/N$ for fixed λ . Why not for λ^*_N ? Is this because λ^*_N is essentially independent from N for these values of N ? Note that in the figure caption you use the notation λ^* whereas in the figure you use just λ . Please correct this.

Reply: For the charge response functions in Figure 4b and 4de, we simply choose various values of λ and we fixed it, in order to establish the error-bar. And as guessed by the referee, λ^* very weakly depends on N for $N > 2$, the values needed for the extrapolation. In the new version of the manuscript we removed these insets, because referee suggested that the figure is too busy, and we rather added $1/\epsilon$ at larger $r_s=4$, to demonstrate the power of the new method.

[DONE]Referee: In the table, no value is given for the polarization for $r_s=4$. Is this because the extrapolation to $N \rightarrow \infty$ becomes simply unreliable? It is stated that no extrapolation for the spin response is needed since CFS and VCCFS approach from below and above, and that therefore the estimate of the spin response is more precise. But looking at the figures, it seems that the spin response just converges faster and is therefore more precise than the charge response. Is this correct? A related more general question: is the calculation limited to $r_s=1-3$? I think it is important to say something about the range in r_s .

Reply: Indeed we found that the spin response function converges much faster than the charge response function, especially at increased values of r_s . For the charge we could not establish the error-bar at $r_s=4$ (even using order $N=6$), while the spin response is reasonably converged even at $r_s=6$ (not shown in the manuscript). However, we wanted to stay on the safe side and showed only data with very high accuracy.

This observation has a physical roots. For 3D UEG, the spin instability first occurs around $r_s \sim 70$, while the charge instability (negative compressibility and divergence of the four-point vertex) develops at $r_s \sim 5.2$. In diagrammatic language, some Feynman diagrams, which are dangerous in the charge channel, are simply absent in the spin channel. This makes the spin response function better behaved than the charge response.

At the end of the Method section, we added a discussion on advantages and limitations of the approach, and we now explain in which regime this method works well (for which r_s).

Referee: How do the authors estimate the final error bars? I think it is important to mention this since the whole point of the paper is to make quantitative and unbiased predictions.

Reply: The systematic error bars of the spin response function are estimated easily, as the CFS scheme converges from below, and VCCFS scheme very rapidly converges and oscillates around the “final converged” value. We take the larger of the two values: the difference between the last two values in the VCCFS scheme, or the distance to the value of the CFS scheme as the extrapolation error.

The systematic error bars for the charge response function are estimated by the extrapolations of different theories and different dressing schemes. We now include in all plots our best estimates for the error-bars.

We also clarify this point in the manuscript (page 3).

[DONE]Referee: The expansion of v_k is taken such that the lowest order in ξ gives the Fock self-energy Σ^x minus $\Sigma^x_{k_F}$. The higher order terms are just shifts in the chemical potential to ensure that the density remains fixed at each order of the diagrammatic expansion. How are these shifts s_N determined in practice? It seems that this requires calculating the density. Because of this choice, Fock diagrams should be excluded. However, there is a Fock diagram at the second order in ξ in Figure 2 of the supplementary material.

Reply: In practice, we calculate the density of the system with $P(q, \tau=0)$ where the limit of the large external momentum $q \gg k_F$ is taken analytically (equivalent to calculating the Green's function).

Numerically, we can think of searching for the root of $P(q, \tau=0) + n = 0$ to tune s_N so that the density does not change order by order.

In practice, we can use a trick: According to our expansion, s_N is of the order ξ^N , and is determined at order $N+1$ so that it cancels contributions to the density from all other diagrams at this order (containing s_{N-1} , ... s_1 or no counter-term).

At order $N+1$ it is easy to see that s_N enters only through the second-order-type diagrams of the form GGG, because s_N already carries very high ξ^N (see the third diagram in the middle row of Fig.2 in the supplement). Then we see that the contribution from s_N is simply $s_N * GGG$, where GGG stands for the value of the lowest-order counter term diagram. The latter is depicted in middle row of Fig.2 in the supplement. We hence do not need to use root finding and change

s_N , but we just compute the value of G , and set s_N to such a value that cancels contribution of all other diagrams at the order $N+1$ to the density.

To answer the second part of the question: with our choice of the self-energy counterterm, any Fock sub-diagrams is exactly cancelled with the corresponding counter-term. We can thus remove Σ^x from the counter-term and Fock diagrams from the expansion, or alternatively, include both and see the cancellation numerically. In practice we do not sample Fock diagrams. We have now clarified the above points in the manuscript and in the supplementary material.

Referee:

- In the caption of figure 2, $n = 2 / (4 \pi r_s^3)$ instead of $n = 3 / (4 \pi r_s^3)$. The Bohr radius is taken to be one here. Please write this somewhere.
- It sounds a bit bizarre to say that RPA is exact for the non-interacting electron gas, since it gives a screened interaction. Maybe it would be better to rephrase this.

We have also addressed all three minor points as the Referee suggested.

Reply to Referee B

Referee: In the manuscript "Feynmann's solution of the quintessential problem in solid state physics", the authors present a novel diagrammatic Monte Carlo (DiagMC) scheme and apply it to the uniform electron gas (UEG). The UEG has long been used as a minimal model of, e.g., electrons in a metal and is of particular importance for the construction of exchange-correlation functionals for density functional theory.

While Diffusion Monte Carlo (DMC) has been very successfully applied to the UEG and has arguably remained the state-of-the-art method for this model since the seminal work by Ceperley and Alder [11], bias introduced by the fixed-node approximation, the need for finite-size extrapolation, and challenges in addressing finite temperature properties or response functions make new accurate and controlled methods highly desirable. The present manuscript presents such a new approach and at least matches the accuracy of DMC predictions for the charge and spin response functions, where available, while not being subject to fixed-node, finite-size, or zero-temperature limitations.

DiagMC denotes a rather recent computational approach to quantum many-body systems based on the stochastic sampling of Feynman diagrams. During the last ten years, it has been applied rather successfully to a wide range of interacting quantum systems, including the unitary Fermi gas, unconventional superconductors, and other correlated lattice models. Fermions with long-range interactions in the continuum, however, which host a certain amount of complications due to short- and long-wavelength divergences, have not been adequately addressed before.

In general, the two fundamental challenges in applying the DiagMC technique are (a) does the diagrammatic series converge sufficiently well to allow for an extrapolation of the calculated finite-order results to the infinite-order limit; and (b) how high a diagram order can be calculated

with sufficiently small stochastic errors before the sign problem becomes too severe. The present work makes major improvements on both fronts.

First, the diagrammatic expansion's starting point is optimized to be close to the emergent Fermi liquid physics of the result. The basic idea of shifting the starting point with an artificial potential has been employed before [Ref. 23 and references therein], but the automatic optimization of these free parameters via the principle of minimal sensitivity and by fixing the Fermi surface volume have not been proposed in this context before, to my knowledge. Second, by grouping similar diagrams into classes whose constituents largely cancel each other, the present work achieves a marked alleviation of the sign problem for given diagram order. Other recent DiagMC works [cf. Ref. 25; also <https://doi.org/10.1103/PhysRevB.97.085117>, <https://arxiv.org/abs/1712.10001>] have explored determinantal schemes that achieve a similar effect by summing all diagrams at a given order, although the computational scheme is quite different in practice.

In total, this work presents a major achievement, which brings the DiagMC technique into the realm of ab-initio simulations for materials. I hence recommend its publication. However, additional details on the method should be provided (possibly in the supplement) and the presentation may be improved in some places, as detailed below.

We thank the Referee for positive evaluation of our work.

Referee: The actual, stochastic, DiagMC sampling process has not been described at all and it is not obvious that the conventional DiagMC procedure as described, e.g. in [Ref. 6], can be applied straightforwardly to the current setup. For example, Hugenholtz and free energy diagrams come with combinatorial prefactors, which are not easily determined in a local update. Also, given the moderate number of free energy diagrams up to sixth order (Fig. 1), one may plausibly perform the sum over topologies deterministically. Hence my request to specify what is actually done. Similarly, for the optimization of λ please specify what quantity is minimized (what susceptibility, at what 4-momentum). A dedicated overview of the computational procedure may also give readers less familiar with DiagMC a better impression of what your algorithm encompasses.

Reply: We thank the Referee for giving a good suggestion to improve the readability of our paper. We have added a dedicated section in the supplementary material to describe the new DiagMC algorithm. Indeed we deterministically sum over all possible diagrams, and use Monte Carlo to sample the internal variables only (time, momenta) (see chapter II in supplementary). The value of λ is optimized by making the physical response function least sensitive to variation of λ at $q=0, \omega=0$. We now explain this point in Fig. 2. When we compute spin-response, we optimize $\chi_s(q=0, \omega=0)$. When we compute charge response, we optimize $P(q=0, \omega=0)$. In practice, the optimized λ s for spin and charge are not very different, hence maybe one could even use the same universal λ . But for now we optimize it for each observable.

Referee: Stochastic error bars should be quantified, and displayed in the figures.

Reply: Thank you for the suggestion. We have done that now everywhere. After presenting results in scaled units (N_F and E_F) the statistical errorbars are clearly noticeable in particular in Fig.3, although they are used also in Figs.2a 2b, 2d. in Figs. 3a & 3b and 4a-d. In Fig.2c are much smaller than the size of the symbol. In Fig.4 we combined the statistical error-bar with error-bar of extrapolation.

[DONE]Referee: According to the authors, their CFS and VCCFS schemes are conserving approximations in the Baym-Kadanoff sense. Wouldn't this require a self-consistent definition of the diagrams in terms of the interacting propagator?

Reply: No, it is not needed. Conventionally the conserving approximations are indeed constructed by the self-consistent propagators and Luttinger-Ward $\Phi[G]$ functional. But this is not necessary.

As long as one follows the Baym-Kadanoff algorithm to construct a diagrammatic series, it automatically satisfies the conservation laws for various important physical quantities. Dressing the interaction line is not a necessary condition, although it is most commonly used in literature.

To follow Baym-Kadanoff proof/construction, one needs to introduce the source term $SU(1,2)$. Next one constructs an approximation for the generating functional $\log Z$ in the bare expansion (not Luttinger-Ward $\Phi[G]$ as in the skeleton self-consistent expansion). Finally, one takes the first order derivative to obtain the single-particle Green's function, and the second-order derivative to get two particle Green's function. As it was proven in the original work by Baym and Kadanoff (see Ref. [1,2] in the supplementary material) the conservation laws will be automatically satisfied even when propagators are bare.

Referee: I am not sure I follow the authors in their discussion of Fig. 4: "The latter [DMC] data are in agreement with our prediction, although the systematic error-bar at any finite q , which are not given in the original papers but can be inferred from the fluctuations of the data points around our smooth curves, is much larger than ours, demonstrating the superiority of the newly developed VDMC method." (That conclusion is also echoed in the abstract.) To some extent, my lack of persuasion may be an issue of the figure, which is rather overloaded and differences between curves are of similar size as line widths and symbol sizes. Furthermore, it is unfortunate that the DMC references do not give error estimates, and the present authors should not repeat the same mistake by not giving error bars for their own data. Lastly, the compared DMC work is over twenty years old, it is fair to assume that they would do better with modern computers and codes. Hence, if the authors do want to show superiority of their method in this particular measure, they should plot data on a scale where differences are meaningful, and provide clear evidence that their errors are significantly smaller than what could be produced with DMC. Otherwise, the conclusions from the comparison should be restricted to consistency with DMC. This does not detract from the new method's additional capabilities compared to DMC, as mentioned above, and the value of being able to check predictions with two completely independent methods (fixed-node, finite-size vs. finite-order, thermodynamic limit).

Reply:

We have changed Fig.4, because the old one was too overloaded, and we added result for $r_s=4$. At $r_s=4$ the comparison to DMC is much more convincing. The DMC data has huge scatter, while our curves, starting with order $N=3$ to $N=\infty$ is fitting the data well, but clearly we can now establish much more precise value for $1/\epsilon$. Similarly, for the spin susceptibility, we can now establish much more precise benchmark results, which are not available in the literature (Table I). We also added error-bars to all our calculations, and in Fig.4 we have both the statistical and error of extrapolation. The statistical error is a very small (a small fraction of the extrapolation error-bar), hence the width of the curves is always larger than our statistical error-bar, and therefore in many curves it is hard to see the error-bar. The extrapolation error-bar was addressed above.

We also slightly modified the wording. We removed the sentence comparing the precision and error-bar of DMC with our method. We now say: "The latter data are in agreement with our prediction, but notice that DMC allows one to calculate only a set of discrete points, while the newly developed VDMC method gives a smooth and very accurate continuous curve, which allows one to resolve the fine structure."

Regarding the statistical error of our method, we also want to point out that each q-point is computed independently of each other, i.e., all points are independent observables in our method, and we do not use any smoothing technique, yet the curves are almost completely smooth.

Referee: Further suggestions for the authors' consideration:

The "DVCCFS" scheme is rarely mentioned and does not seem to add much to the manuscript. Cutting it from the main text and figures may streamline the presentation.

Reply: In the new version of the manuscript, we now use only two schemes, CFS and VCCFS. The third scheme is removed for clarity, as suggested by the referee.

Referee: In Figs. 2(a, b) and 4(a), it would make sense to indicate the position of λ_N on each curve.

Reply: The figure is pretty busy now, and our attempt at adding arrows in the figure made it worse, so we decided to abandon this idea.

[DONE]*Referee:* Fig. 3(b): Systematic and/or stochastic errors apparently become a lot more pronounced at large momenta. I wonder if the authors have any insight why that is the case?

Reply: It is actually related to the definition of the local-field correction, $G(q)=q^2/8\pi^*(\chi_{-1}^{\text{free}}-\chi_{-1}(q,\omega=0))$. At large momenta $q \gg k_F$, the error bars are dramatically amplified by a huge factor $(q/k_F)^2$. But this is only in the local-field correction quantity, not in physical response functions.

Referee: Fig. 4: As hinted above already, this figure is completely overloaded and it is plainly impossible to discern all the different curves. I would suggest to separate convergence plots - observable vs order at selected k points - from plots showing the k -dependence - extrapolated observable (only, or possibly + one finite-order result) with error bars vs k .

Reply: We followed the advice of the referee, and we now only plot the observables $1/\epsilon$ (with error-bars) for four different values of r_s , including $r_s=4$, where the comparison to DMC is more revealing.

[DONE]Referee: It would be helpful to state a concise definition of the UEG model somewhere in the main text or methods section, say, by stating the Hamiltonian and units. While specialists can certainly defer these from the Lagrangian, it is unnecessarily complicated by the HS transform and variational parameters.

Reply: We thank the Referee for the good suggestion. We have added a dedicated paragraph in the method section to introduce the UEG model, and we start from UEG Hamiltonian.

[DONE]Referee: I am afraid the use of the term "sign blessing" in the present work, while admittedly catchy, may further compound the existing confusion around this expression. To my knowledge, the term was coined by Prokofev and Svistunov to point out that the fermionic sign can make a diagrammatic series convergent due to cancellation between terms with different sign at high orders. In that sense, "sign blessing" refers to a property of the series and is largely orthogonal to the sign problem. The "sign-blessed" diagram groups in the present context, in contrast, leave the series and its convergence properties unchanged but alleviate the sign problem by reducing the variance of the sampled distribution.

Reply: We are trying to be very careful with our term "sign blessed groups", which is distinct from the "sign blessing phenomena", proposed by Nikolay and Boris, as nicely summarized by the referee. We do not have a nicer/better term for this. But notice that in "sign blessing" and "sign blessed groups" have a common idea, namely, a massive cancelation of Feynman diagrams due to fermionic statistics.

We added a section in the supplementary material, to make this point more clear (Section II and III).

Referee:

- The "VCCFS" scheme seems to be sometimes called "LVCFS". This inconsistency should be fixed.
- In the last paragraph of p. 2, the reference should be to Fig. 3, rather than 1, of the supplement. "Hartree" is misspelled "Hatree" a couple of times.
- Below Fig. 4: The $\$V_q\$$ appearing in the definition of the polarization function is not defined anywhere, as far as I can see.

Reply: We have also addressed all other minor points as the Referee suggested.

Reply to Referee C

Referee: The manuscript of Chen and Haule presents a new diagrammatic Monte Carlo method, building considerably on the work of a number of researchers in the last few years (of which I am not one). The work presents some impressive and convincing results in the case of the uniform electron gas in the thermodynamic limit – a significant development from previous (lattice) results, with the inclusion of a true long-range Coulomb interaction, and the computational of important two-particle quantities. The nice features of the approach include:

- Screened interaction in the zeroth-order terms, with optimized screening length
- Grouping of diagrams to control sign problem (though I strongly dislike the term ‘sign-blessed’!)
- Two approaches to combine diagrams summed to infinity analytically
- Continuous momentum / imaginary time resolution
- Significant deviation from RPA-like behaviour found, even though there is essentially broad agreement with DMC.

The results are impressive in the accuracy they achieve for this paradigmatic model, and the manuscript is certainly worthy of careful consideration in this journal due to its important contribution to the field, and opening of a new avenue within this field. However, I have a number of comments which should be addressed before publication – some of a more general type, and others of a specific technical nature.

We thank the Referee for positive evaluation of our work.

Referee: The scope, drawbacks and limitations of the approach are not discussed (and therefore solely defined by what is not in the manuscript). More consideration should be given to the areas in which the method can particularly excel, and why. For instance, it is known that currently for lattice models, diagrammatic Monte Carlo is only predictive up to $U \sim 4t$ – presumably this carries over to r_s . This is despite the high- r_s limit (Wigner crystal) in some sense being easier, with the effective interaction shorter ranged. Presumably the limitation is that the convergence with N becomes slower as r_s increases? What is the limitation in going to higher r_s ? How does the explicit inclusion of a screened interaction in L_0 improve this? Furthermore, some discussion should be taken by the scaling of the computational effort both with N and r_s .

Reply: We have added a paragraph in the Method section to summarize our contribution and discuss the advantages and limitations of our method. We also summarize it here: In general, we do not expect the simple variational theory can work at large r_s (beyond $r_s \sim 6$ for spin and $r_s \sim 4$ for charge), where the system approaches an instability in the charge channel. Beyond this r_s , some sophisticated vertex renormalizations, more than a static screening constant (presumably, certain four-point vertex functions), have to be introduced into the variational theory, to control the diagrammatic series. We have to admit that the high- r_s limit (Wigner crystal) is beyond the ability of our current approach, mainly because we do not know what are the relevant vertex functions that have to be renormalized.

In terms of our achievements of the method, and in what way the new method really excel compared to the old techniques include:

- a) Minimizing the variance of the weight function to reduce the statistical errors.
- b) Improving the efficiency of the Monte Carlo updates by grouping diagrams.

- a) In the supplementary material we added a section, where we discuss three different weight functions, F_1 , F_2 and F_3 . F_1 is the physical answer we are looking for, F_2 is the weight function that our Monte Carlo method samples, and F_3 is the weight function that the conventional diagrammatic MC samples. We expect that F_3 diverges factorially with N , while F_1 is negligibly small compared to F_3 . Currently we do not know the scaling of F_2 , but we expect that it can vary all the way from exponential to factorial, depending on the problem, and the choice of basis. Our current choice of basis guarantees that $F_2 \ll F_3$.
- b) When it comes to improving efficiency of the updates, we can definitely claim exponential improvement as compared to conventional DiagMC, as we are grouping 2^N diagrams into Hugenholts-type diagrams. On top of that, many propagators in the sign-blessed groups are shared, hence we evaluate them simultaneously, further improving the efficiency.

We added this discussion in the supplementary material.

Referee: It is well known certain diagrammatic expansions are only conditionally convergent, and that along with the ‘pillars’ being the quality of the starting point, and the ability to calculate contributions to high enough order, the other key question is the uniqueness of the result, and the assurance that the summation of diagrams will indeed converge via finite summation. What assurances are there of this for the diagrams which are summed, and the computational algorithm employed here?

Reply: In our diagrammatic expansions, we do not rely on self-consistent Dyson equations, which cause the misleading convergence problem for the skeleton diagrammatic series studied before (dressing of the series). All schemes we have been using are a standard Taylor power series in ξ , the uniqueness of the limit (if the convergence occurs) is mathematically assured. But the convergence might not be there, as it happens for charge response beyond $r_s \geq 4$ and spin beyond $r_s \geq 6$.

Referee: The first paragraph talks about the method being able to calculate the ‘momentum-frequency’ resolved response functions. However, I see no frequency resolution in any of the results presented, as all the results are simply giving the static limit. Why is no frequency-dependence given, as its omission looks like a limitation of the method. I am confused as to how the $w=0$ results were obtained even. The diagrams are sampled in τ -space, therefore presumably you are continuing the result to real-frequency (or integrating to get the static limit)? Is the lack of frequency resolution down to the uncertainty in the continuation, or the some other limitation which I am missing? All the results obtained could instead be computed just by looking at the results of static perturbations (as was done for the DMC comparisons). This is an important point which must be addressed, to determine the viability of the method for the true dynamics of the system.

Reply: The Monte Carlo method we propose can calculate full k - τ dependent response functions, which allows us to use numerical analytical continuation protocols to extract momentum-frequency resolved response functions. Hence, there is no limitation in obtaining frequency response function. However, for the space-constraints of this paper, we did not include any frequency-dependent object here, although we actually compute them.

The $\omega=0$ part of a bosonic response function is equal to the $i\omega=0$ response, and is simply integral over all imaginary times, hence no continuation or extrapolation is needed. We have the entire τ -dependent quantities, but we do not show them here.

Referee: Continuing on from the point above, surely the λ^* values would depend on both the momentum and frequency point considered. This would be especially true of changing frequency point, as (I presume) you would expect the λ^* value to tend to 0 in the high-frequency limit? If the aim is to get the full dynamics in one calculation, then this would not be possible?

Reply: In the high frequency or large momentum limit, we essentially probe the bare vertex functions (which is the bare electron propagator), meaning that the diagrammatic series converge extremely fast (say with order one or two). Since the first two orders do not contain counter-term, the high-frequency limits is very precise for almost any λ , and there is no need to fine tune λ to obtain the high-frequency/momentum part.

This is also consistent with principal of minimal sensitivity: more weakly correlated regime tends to be more insensitive to the change of auxiliary parameters.

However, it might be slightly more efficient to allow λ to be somewhat momentum dependent, and we tested that scheme, but decided that it does not sufficiently improve the results to warrant making the algorithm more complicated and adding more variational degrees of freedom.

Referee: Furthermore, the optimization of λ is performed with a ‘minimal sensitivity’ principle. In effect, this is looking for a number of derivatives minimized in the variation of λ wrt the observable. However, it is seen that this is far from clear for lower values of N in figure 2, where there is a strong N dependence, and arguably for $N=1,2,3$, the least sensitive value of λ is at larger values than even plotted. Can the authors comment on this, and the sensitivity of λ^* wrt N (as well as momentum and frequency as discussed above).

Reply: Indeed the optimal λ at order=1 is infinity, while for any $N>1$ there is a clear extremum (as shown in the plots). From simple analysis of limits, we know that at large λ all orders should saturate to a constant, and at small λ and order ≥ 2 has to diverge (and oscillate). At infinite order all λ s should give the same correct result. Numerically we find extremum for all order $N\geq 2$ (actually maximum) at finite λ . However, $\lambda=\infty$ is not problematic (or unphysical) at $N=1$, as this just gives regular RPA result, which is known to be the best at the leading order in perturbation theory.

Referee: It is implied that for χ_s that the LVCFS and CFS bracket the $N \rightarrow \infty$ result for any finite N . This is presumably not proved, and while the numerical results seem to indicate this, it is not a rigorous result which can be relied on (as seen for the charge response).

Reply: The Referee is right that the different trends of two schemes are not mathematically proven. We do not use this anymore in the manuscript.

Referee: Similarly, there is presumably also no guarantee of monotonicity in the results, despite the groupings of diagrams and numerical evidence (this also somewhat relates to the formal convergence of the approach).

Reply: The Referee is correct that the monotonicity is based on numerical evidence. We have pointed this out explicitly in the discussions of the error bar estimations for the spin response function.

Referee: Can something be explicitly said about whether all independent time-orderings of a particular diagram are included in a group, or whether these are considered in distinct groups – I was unclear on this from the manuscript.

Reply: In our approach the time variables are sampled by Monte Carlo, and we did not try to order times. All contributions to Baym-Kadanoff conserving group are however included in a group, which we found to be very important.

We checked that time symmetrization slightly (approximately factor of 2) improves the variance of the weight function. This means that every Hugenholtz diagram is being used, not just the topologically distinct once (see discussion above). So that all possible exchanges of times are within the same group. We have implemented two schemes, one without the symmetrization and one with symmetrization, and we use both. While symmetrization reduces the error-bar, it is also a bit more expensive (10%), overall still a bit better than without symmetrization.

Referee: I was very confused at parts by the acronyms. DVCCFS is defined, but never seemingly used, while later on in the manuscript, LVCFS crops up, but is never defined. Is this latter acronym meant to be VCCFS? If DVCCFS is not used, then it should be removed. Furthermore, above Table I, a 'VDMC' acronym is used, and not defined?! Presumably this is meant to be LVCFS??

Reply: We thank the referee for pointing out these issues. LVCFS==VCCFS, and we corrected the mistake. VDMC is Variational Diagrammatic Monte Carlo, which we now spell out. DVCCFS was used in the inset of Fig.4 but is now abandoned in the updated version.

Referee: I am confused by the assertion that 'Contrary to the finite momentum response, the static charge response $P(q=0, w=0)$ can be obtained from the ground state energy of the system'. Surely this is not true, and the momentum-resolution can also be obtained (see for instance the reference 31), where the momentum resolution is obtained. Furthermore, momentum resolved DMC results are given in Fig 4 anyway?!

Reply: We thank the Referee for pointing this out. We have made our sentence more accurate: “Contrary to the finite momentum response, the static charge response $\chi(q=0, \omega=0)$ can be obtained from the ground state energy of the system without explicitly introducing a modulated external potential.”

Referee: Figure 4 for has too many plots and legends to work out what is going on in a reasonable way, with the detail too small to be presentable. I am sympathetic to space constraints, but this is too compact.

Reply:

Yes, as also suggested by referee B, we have completely changed Fig.4 so that it is more clear, and gives more relevant comparison with DMC at $r_s=4$.

Referee:

- ‘Feynman’ is incorrectly spelt in the title of the manuscript (!)
- line 5: missing ‘a’
- ‘ P_q with the increasing order The inset of Fig. 4b’ (missing full stop)

Thank you. We corrected all minor points.

List of changes:

1. For better presentation, we decided to use dimensionless quantities throughout the paper. In particular, the spin-susceptibility and the polarization function are now divided by the density of states at the Fermi level (N_F), so that the non-interacting result has $\chi^0/N_F=1$ and $P_{RPA}(q=0, \omega=0)/N_F=1$. We similarly scaled λ with E_F , and q with k_F . In this way, the spin-susceptibility in Fig.2 is now clearly increasing with increasing r_s , as expected for more correlated system, rather than decreasing due to the rapid decrease of the density. Also the numerical (statistical) noise is now more clear in Fig.2, as before it was hidden in small values of χ at $r_s=4$.
2. We added error-bars in all calculated data and figures, as suggested by the referees.
3. In Fig2.b we now compute the local-field correction by subtracting the RPA result at the same temperature as the MC data are obtained, while before we subtracted the zero temperature RPA results, as the usual definition of the local-field correction commonly involves $T=0$ RPA result. By using the same temperature in simulation and in RPA, we reduced the effect of finite temperature in our simulations. Namely, the Monte Carlo data are computed at low but finite temperature $T=0.04E_F$ and it is more appropriate to redefine the local field correction so that one subtracts the RPA results at the same low temperature. This now more clearly shows the emergence of $2*k_F$ peak in local field correction (Fig.3b), and it slightly modifies the shape of the $2*k_F$ peak.
4. Fig.4 was too busy, as pointed out by the referees. We replaced panel a), which contained less important information on λ dependence (similar to Fig.2a) by the computed dielectric function for $r_s=4$, where comparison to DMC is possible, and clearly shows

usefulness of our more precise calculation to establish more precise benchmark for the future.

5. Finally we note that we found a small numerical issue in how spin-susceptibility was computed in our original version of the manuscript (but not charge). By developing a completely new independent code, which uses slightly different algorithm in generating diagrams (and different way of sampling them), we spotted the error in the original code, and corrected it in this version of the manuscript. We now have two completely independent codes using different variations of the described algorithm, which give the same result within the statistical error, so that we are confident that the results are now ergodic and bug-free.
6. We also added comparison of our spin susceptibilities to the estimates from the available literature (column 2 in Table 1). Our values agree with the existing literatures, but are much more precise than what is currently available.
7. We now clarify the benefits of grouping diagrams in the second paragraph on the page 2.
8. We have corrected the issues with coupling constants in Eq. (1), (2) in the manuscript, and Eq. (6), (7) in the supplementary material, as our original choice of coupling constants was unconventional.
9. We changed “so that with the increasing expansion order the susceptibility converges towards the exact result” in page 3 to “so that all theories describe the same UEG in the infinite order limit.” We also modified the sentence “When the MPS is used... the convergence to the exact results is very rapid ... ” to make it clear that the convergence of the response functions are based on numerical evidence, as suggested by the referee.
10. In the caption of figure 2 corrected $n = 3 / (4 \pi r_s^3)$.
11. We now explain in the method section that the units used in the simulations are Rydbergs (the atomic Rydberg units $\hbar=2m=e^2/2=1$).
12. We made the sentence “Contrary to the finite momentum response, the static charge response $P(q=0, \omega=0)$ can be obtained from the ground state energy of the system without explicitly breaking the spin symmetry using an external magnet field” more accurate.
13. We added a new reference by R. Rossi, Phys. Rev. Lett 119, 045701 (2017).
14. We now explain in the method section how the chemical shifts are determined by calculating the density of the system.
15. We added a paragraph in the Method section to introduce the Hamiltonian of the UEG model.
16. We added two paragraphs in the Method section to discuss the advantages and limitations of our newly developed method, as suggested by the referee.
17. We clarify the protocol to generate the diagrams and the corresponding basis in the caption of Fig. 1, and in more detail in supplementary information.
18. We now explain why the Fock sub-diagrams exactly cancel with the counter-term in the caption of Fig. 2 and in the supplementary material.
19. We wrote a dedicated section in the supplementary material to explain the new diagrammatic Monte Carlo algorithm we have used to calculate the physical quantities in this paper.
20. We corrected several minor points and typos.

REVIEWERS' COMMENTS:

Reviewer #1 (Remarks to the Author):

The authors have answered my questions in a clear way and improved the manuscript accordingly. I recommend publication in Nature Communications.

I still have a few minor suggestions for the revised version, which in my opinion will make it more clear. While reading the paper I also found a number of simple typos.

-The differences with existing diagrammatic Monte Carlo algorithms are now explained in the conclusion. In the introduction, however, the authors write "Our solution employs a recently developed diagrammatic Monte Carlo algorithm,..". I think the word "algorithm" is misleading, since the new algorithm is very different from conventional diagrammatic Monte Carlo. The word "approach" or "method" might therefore be more suited. Moreover, I find it strange that Refs. 6 and 7 are excluded here, for some reason.

-The authors write "The previously used diagrammatic Monte Carlo algorithms, which were sampling the diagrams one by one, are highly inefficient here." This ignores the existence of the CDet algorithm by R. Rossi. Therefore I would suggest to write: "The *_conventional_* diagrammatic MC algorithms *_[add refs. here: 3-8]_*, which were sampling the diagrams one by one ..."

-The authors have made the caption of Figure 1 more clear, but I still have one objection: In figure 1, they choose to keep only topologically distinct diagrams, which corresponds to their approach a) as explained in the answer to my report. It is therefore weird to state at the end: "the above protocol can generate multiple copies of the same Feynman diagram, which we weight with a proper symmetry factor". This does not make much sense without knowing that "We tested two approaches: a) we just keep one copy to avoid double counting. b) we keep all copies and precompute the symmetry factors." It would be good to add this at the very end of the caption.

-The blue shaded area in Figure 2, panel d, seems to have no purpose. It was confusing in combination with the sentence "The shaded region..". I would remove the blue area in panel d.

Some typos:

P1. "The effects of the interaction is included.." -> "The effect.."

P1. "We take the screening parameter as variational parameters" -> "parameter"

P2. Figure 1: there is a problem with momentum conservation in the green panel: one k_1 should be k_2 .

P5. "The density ρ is $\rho_q = \dots$ ": there is a q that should be bold.

P5. "intermediate"

Supplementary P3: "landscape"

Reviewer #2 (Remarks to the Author):

The authors have addressed my earlier comments and the revised presentation is much clearer. I recommend its publication.

Reviewer #3 (Remarks to the Author):

Previous concerns about the manuscript have been satisfactorily answered, and I am happy for publication.

Reply to Referee #1

Referee: The authors have answered my questions in a clear way and improved the manuscript accordingly. I recommend publication in Nature Communications.

I still have a few minor suggestions for the revised version, which in my opinion will make it clearer. While reading the paper I also found a number of simple typos.

-The differences with existing diagrammatic Monte Carlo algorithms are now explained in the conclusion. In the introduction, however, the authors write “Our solution employs a recently developed diagrammatic Monte Carlo algorithm,..”.

I think the word “algorithm” is misleading, since the new algorithm is very different from conventional diagrammatic Monte Carlo.

The word “approach” or “method” might therefore be more suited. Moreover, I find it strange that Refs. 6 and 7 are excluded here, for some reason.

Reply: We changed the "algorithm" to "method" and added Ref.6&7 at this point as well.

Referee: The authors write “The previously used diagrammatic Monte Carlo algorithms, which were sampling the diagrams one by one, are highly inefficient here.”

This ignores the existence of the CDet algorithm by R. Rossi. Therefore I would suggest to write: "The *_conventional_* diagrammatic MC algorithms *_[add refs. here: 3-8]*_, which were sampling the diagrams one by one ..."

Reply: We have changed the sentence as suggested by the referee.

Referee: The authors have made the caption of Figure 1 more clear, but I still have one objection:

In figure 1, they choose to keep only topologically distinct diagrams, which corresponds to their approach a) as explained in the answer to my report.

It is therefore weird to state at the end: “the above protocol can generate multiple copies of the same Feynman diagram, which we weight with a proper symmetry factor”.

This does not make much sense without knowing that "We tested two approaches:

a) we just keep one copy to avoid double counting.

b) we keep all copies and precompute the symmetry factors.”

It would be good to add this at the very end of the caption.

Reply:

We corrected the caption, as suggested by the referee.

Referee: The blue shaded area in Figure 2, panel d, seems to have no purpose. It was confusing in combination with the sentence “The shaded region..”.

I would remove the blue area in panel d.

Reply: We want to emphasize the effect of interaction, hence the shading.

Referee: Some typos:

P1. "The effects of the interaction is included.." -> "The effect.."

P1. "We take the screening parameter as variational parameters" -> "parameter"

P2. Figure 1: there is a problem with momentum conservation in the green panel: one k_1 should be k_2 .

P5. "The density ρ is $\rho_q = \dots$ ": there is a q that should be bold.

P5. "intermediate"

Supplementary P3: "landscape"

Reply: We corrected all. Thank you.

Reply to Referee #2

Referee: The authors have addressed my earlier comments and the revised presentation is much clearer. I recommend its publication.

We thank the Referee for his careful reading of the manuscript and his suggestion.

Reply to Referee #3

Referee: Previous concerns about the manuscript have been satisfactorily answered, and I am happy for publication.

We thank the Referee for his positive response and his recommendation.